# Impact of *miR-1*/*miR-133* Clustered miRNAs: *PFN2* Facilitates Malignant Phenotypes in Head and Neck Squamous Cell Carcinoma

**DOI:** 10.3390/biomedicines10030663

**Published:** 2022-03-12

**Authors:** Shunichi Asai, Ayaka Koma, Nijiro Nohata, Takashi Kinoshita, Naoko Kikkawa, Mayuko Kato, Chikashi Minemura, Katsuhiro Uzawa, Toyoyuki Hanazawa, Naohiko Seki

**Affiliations:** 1Department of Functional Genomics, Chiba University Graduate School of Medicine, Chiba 260-8670, Japan; cada5015@chiba-u.jp (S.A.); naoko-k@hospital.chiba-u.jp (N.K.); mayukokato@chiba-u.jp (M.K.); 2Department of Otorhinolaryngology/Head and Neck Surgery, Chiba University Graduate School of Medicine, Chiba 260-8670, Japan; t.kinoshita@chiba-u.jp (T.K.); thanazawa@faculty.chiba-u.jp (T.H.); 3Department of Oral Science, Graduate School of Medicine, Chiba University, Chiba 260-8670, Japan; axna4812@chiba-u.jp (A.K.); minemura@chiba-u.jp (C.M.); uzawak@faculty.chiba-u.jp (K.U.); 4MSD K.K., Tokyo 102-8667, Japan; nijiro.nohata@merck.com

**Keywords:** microRNA, clustered, HNSCC, *miR-1-3p*, *miR-206*, *miR-133a-3p*, *miR-133b*, TCGA, *PFN2*

## Abstract

Based on our original RNA sequence-based microRNA (miRNA) signatures of head and neck squamous cell carcinoma (HNSCC), it was revealed that the expression levels of *miR-1-3p*, *miR-206*, *miR-133a-3p*, and *miR-133b* were significantly suppressed in cancer specimens. Seed sequences of *miR-1-3p*/*miR-206* and *miR-133a-3p*/*miR-133b* are identical. Interestingly, *miR-1-3p*/*miR-133a-3p* and *miR-206*/*miR-133b* are clustered in the human genome. We hypothesized that the genes coordinately controlled by these miRNAs are closely involved in the malignant transformation of HNSCC. Our in silico analysis identified a total of 28 genes that had putative *miR-1-3p*/*miR-133a-3p* and *miR-206*/*miR-133b* binding sites. Moreover, their expression levels were upregulated in HNSCC tissues. Multivariate Cox regression analyses showed that expression of *PFN2* and *PSEN1* were independent prognostic factors for patients with HNSCC (*p* < 0.05). Notably, four miRNAs (i.e., *miR-1-3p, miR-206, miR-133a-3p*, and *miR-133b*) directly bound the 3′untranslated region of *PFN2* and controlled expression of the gene in HNSCC cells. Overexpression of *PFN2* was confirmed in clinical specimens, and its aberrant expression facilitated cancer cell migration and invasion abilities. Our miRNA-based strategy continues to uncover novel genes closely involved in the oncogenesis of HNSCC.

## 1. Introduction

Head and neck squamous cell carcinoma (HNSCC) is a malignant neoplasm that arises mainly from the mucosa of the oral cavity, pharynx, and larynx [1]. HNSCC is the sixth most common cancer worldwide, with 890,000 new cases and 450,000 deaths in 2018 [2]. Epidemiological studies have shown several risk factors for HNSCC such as consumption of tobacco and alcohol, exposure to environmental pollutants, and infection with human papillomavirus or Epstein–Barr virus [3]. With an increase in the number of HPV-related HNSCC with favorable prognosis, the overall survival rate of HNSCC is improving [4]. However, the prognosis of HPV-negative HNSCC has not improved, even with multidisciplinary treatments combining surgery, irradiation, chemotherapy, molecular targeted agents, and immunotherapy [3,5]. More than 60% of HNSCC cases are at an advanced stage at the time of the first diagnosis [1]. Treatment that combines chemotherapy with radiation therapy or surgical resection is the first option for locally advanced HNSCC patients [6]. Due to the anatomical characteristics of HNSCC, these treatments have a significant impact on the quality of life. Despite the use of invasive procedures, there are many cases resistant to treatment and recurrence or metastasis after treatment is not uncommon [7]. To further improve treatment outcomes, it is necessary to elucidate the molecular biological mechanisms that underlie recurrence and metastasis of HNSCC.

MicroRNA (miRNA) is a type of single-strand, noncoding RNA, and its length is only 18–25 nucleotides [8]. miRNA acts as a negative controller of gene expression in a sequence-dependent manner [8,9]. In human cells, a single species of miRNA can control a vast number of genes, and the expression of a single mRNA is subject to numerous miRNAs [8,10]. Bioinformatic analysis suggested that more than 60% of protein-coding genes are controlled by miRNAs [8,11]. Therefore, aberrant expression of miRNAs likely disrupts intracellular RNA networks. In fact, numerous studies have demonstrated that aberrantly expressed miRNAs are involved in human diseases including various types of cancer [12,13].

Recent advances in nanotechnology have led to the development of drug delivery systems that deliver various drugs to target cancer cells [14]. Many attempts to use miRNAs as pharmaceuticals have been reported so far [15,16]. The advantage of miRNAs as drugs is that one type of miRNA has the potential to control many target genes [15]. Recently, exosomes have been attracting attention as a drug delivery system. Exosomes are a type of cell-derived vesicle characterized as extracellular vesicles. Of particular note, some miRNAs are contained within exosomes and migrate between cells through exosomes [14,17]. The development of new therapies that embed tumor-suppressive miRNAs in exosomes and deliver them to cancer cells is very attractive.

Interestingly, some miRNAs are in close proximity within the human genome. These miRNAs are called clustered miRNAs [18,19]. The clustered miRNAs share a number of properties: (a) they are composed of physically adjacent miRNA genes that are transcribed together in the same orientation, (b) no separate transcriptional units exist between the members of the cluster, and (c) there are no miRNAs in opposite directions [19]. Members of miRNA clusters have been shown to exhibit similar expression levels and often regulate genes and biological functions belonging to the same signaling pathway [20]. Clustered miRNAs work more efficiently than single miRNA genes, because they include numerous miRNA-coding genes [19]. Detailed analysis of clustered miRNAs will be an important topic in future miRNA research.

We have created several miRNA expression signatures of HNSCC that originated in several regions, e.g., maxillary sinus, oral cavity, and hypopharynx [21,22,23]. Analysis of our miRNA signatures of HNSCC revealed that the expression levels of *miR-1-3p, miR-206*, *miR-133a-3p*, and *miR-133b* were significantly downregulated in cancer tissues. Interestingly, *miR-1-1-3p*/*miR-133a-2*, *miR-133a-1*/*miR-1-2-3p*, and *miR-206*/*miR-133b* are clustered miRNAs in the human genome, specifically, 20q13.33, 18q11.2, and 6p12, respectively [24]. Moreover, the seed sequences of *miR-1-3p*/*miR-206* and *miR-133a-3p*/*miR-133b* are identical [25]. We hypothesized that searching for genes/molecular pathways commonly controlled by these clustered miRNAs would enhance our understanding of the molecular pathogenesis of HNSCC.

Our analysis revealed that Profilin 2 (*PFN2*) was directly controlled by *miR-1-3p, miR-206*, *miR-133a-3p*, and *miR-133b*, and its expression was involved in HNSCC pathogenesis. Functional studies demonstrated that aberrant expression of *PFN2* facilitated the migratory and invasive abilities in HNSCC cells. Searching for antitumor miRNAs and the target molecules that these miRNAs coordinately control will improve our understanding of HNSCC.

## 2. Materials and Methods

### 2.1. Analysis of miRNA Expression in HNSCC

Expression levels of each miRNA in HNSCC clinical specimens was examined based on the miRNA expression signature (GSE184991) and TCGA–HNSC data (TCGA, Firehose Legacy).

Six cDNA libraries obtained from three paired of cancer and normal tissues were sequenced by Next Seq500 (Illumina, San Diego, CA, USA) for miRNA expression signature. The clinical information of three HNSCC patients used for miRNA sequencing are shown in Appendix A. All specimens used for our miRNA signature were derived from surgical resection at Chiba University Hospital. Cancer tissues were collected from each primary tumor, and normal tissues were collected from normal mucosa at least 1 cm away from the margins of the primary tumor.

The TCGA–HNSC miRNA sequence expression data were downloaded from cBioportal (https://www.cbioportal.org), accessed on 10 April 2020 [26,27].

### 2.2. HNSCC Cell Lines and Cell Culture

SAS and Sa3 were purchased originally from the RIKEN BioResource Center (Tsukuba, Ibaraki, Japan). These cell lines were cultured in DMEM medium with 10% fetal bovine serum and antibiotics (i.e., penicillin/streptomycin) The cells were grown in a humidified atmosphere of 5% CO_2_ and 95% air at 37 °C. The features of the cell lines are shown in Appendix A.

### 2.3. Transfection of Mature miRNAs and siRNAs

The protocol used for transient transfection of miRNAs and siRNAs were described in our previous studies [22,28,29]. All miRNA precursors were transfected at 10 nM, and siRNAs were transfected at 5nM into HNSCC cell lines using RNAiMAX (Invitrogen, Carlsbad, CA, USA). Mock was a group without precursors or siRNAs. Control groups were transfected with the negative control precursor. The reagents used in the analysis are shown in Appendix A.

### 2.4. Functional Assays (Cell Proliferation, Migration, and Invasion) Conducted in HNSCC Cells

The procedures for functional assays (cell proliferation, migration, and invasion assays) in HNSCC cells have been described previously [22,28,29]. Briefly, in proliferation assay, SAS or Sa3 cells were plated at 3.0 × 10^3^ cells per well in 96-well plates. Cell proliferation was examined by XTT assays (Sigma–Aldrich, St. Louis, MO, USA) 72 h after miRNA/siRNA transfection. For migration and invasion assays, SAS or Sa3 cells at 2.5 × 10^5^ cells per well were transfected in 6-well plates. After 48 h transfection, SAS or Sa3 at 1.0 × 10^5^ cell per well were added into the Corning BioCoat^TM^ cell culture chamber (Corning, Corning, NY, USA) for migration assays or into the Corning BioCoat Matrigel Invasion Chamber for invasion assays. After 48 h, the cells at the bottom of the chamber were counted and analyzed.

### 2.5. Identification of Putative Targets Controlled by miR-1/miR-133 Clustered miRNAs in HNSCC Cells

The seed sequences of *miR-133a*/*miR-133b* and *miR-1-3p*/*miR-206* were confirmed based on miRbase v.22.1 (https://www.mirbase.org, accessed on 10 April 2020) [30].

We selected putative target genes that had both *miR-133a*/*miR-133b-* and *miR-1-3p*/*miR-206*-binding sites based on TargetScanHuman v.7.2 (http://www.targetscan.org/vert_72/; data downloaded on 10 July 2020) [31]. The clinicopathological analysis of candidate genes were performed using clinical information of TCGA–HNSC obtained from cBioportal (https://www.cbioportal.org), accessed on 10 April 2020 [26,27].

Five-year overall survival rates between the groups were analyzed by log-rank test. In addition, the multivariate statistical technique was performed using Cox’s proportional hazards model. The ç cases were divided into two groups according to the median value of each gene in OncoLnc (http://www.oncolnc.org; accessed on 20 April 2021) [32].

### 2.6. RNA Extraction and Quantitative Reverse-Transcription PCR (qRT-PCR)

Total RNA was isolated using TRIzol reagent and the PureLink™ RNA Mini Kit (Invitrogen/Thermo Fisher Scientific (Waltham, MA, USA)). Reverse transcription was achieved with the High Capacity cDNA Reverse Transcription Kit (Applied Biosystems, Waltham, MA, USA). We performed qRT-PCR using the StepOnePlus™ Real-Time PCR System (Applied Biosystems). *GAPDH* was used as the normalized control. Taqman assays (Applied Biosystems) used in this report are shown in Appendix A.

### 2.7. Western Blotting

The procedures for Western blotting have been described previously [22,29,33]. We incubated the membranes with the anti-PFN2 antibody (1:500) overnight at 4 °C and with the secondary antibody for 1 h at room temperature. GAPDH was used as an internal control. The reagents used in the analysis are shown in Appendix A. Full blots are shown in Appendix A.

### 2.8. Immunostaining

Paraffin sections were obtained from HNSCC cases who received surgical treatment at Chiba University Hospital. The clinical features are shown in Appendix A. Specimens were incubated with anti-PFN2 antibody (1:1000) overnight at 4 °C. We incubated samples with secondary antibody for 30 min at room temperature and counterstained them with hematoxylin. The reagents used in the analysis are shown in Appendix A.

### 2.9. Dual Luciferase Reporter Assays

*PFN2* DNA sequences including or lacking predicted miRNA-binding sequence were inserted into the psiCHECK-2 vector (C8021; Promega, Madison, WI, USA). Transfection of the purified plasmid vectors into HNSCC cells were performed using Lipofectamine 2000 (Invitrogen) at 50 ng/well. After 48 h of transfection, we conducted dual luciferase reporter assays using the Dual Luciferase Reporter Assay System (Promega). Luminescence data are presented as the Renilla/Firefly luciferase activity ratio.

### 2.10. Gene Set Enrichment Analysis (GSEA)

To investigate the molecular pathways in HNSCC, GSEA was performed. TCGA–HNSC data were divided into high- and low-expression groups according to the Z-score of the *PFN2* expression level. We generated a ranked list of genes by the log_2_ ratio comparing the expression levels of each gene between the two groups. We uploaded the resultant gene lists into GSEA software [34,35] and applied the Hallmark gene set in The Molecular Signatures Database [34,36].

### 2.11. Statistical Analysis

JMP Pro 15 (SAS Institute Inc., Cary, NC, USA) was used for statistical analyses. Comparisons between the two groups were assessed by Welch’s *t*-test. Differences between multiple groups were assessed by Dunnett’s test compared to control group. A *p*-value < 0.05 was considered statistically significant. Significant differences within the figures are expressed as follows: * *p* < 0.05, ** *p* < 0.01, *** *p* < 0.001, N.S.: not significant. Quantitative data are presented as the means and standard errors.

## 3. Results

### 3.1. Expression Levels of miR-1/miR-133 Clustered miRNAs in HNSCC Clinical Specimens

In the human genome, *miR-1-1* and *miR-133a-2* are located on chromosome 20q13.33, whereas *miR-133a-1* and *miR-1-2* are located on chromosome 18q11.2, while *miR-206* and *miR-133b* are located on chromosome 6p12.2 (Figure 1A). Throughout the maturation process, *miR-133a-3p*/*miR-133b* and *miR-1-3p*/*miR-206* are formed from pre-miRNAs. The seed sequences of *miR-133a-3p*/*miR-133b* are identical and the seed sequences of *miR-1-3p*/*miR-206* are identical (Figure 1A).

To confirm the expression levels of *miR-133a-3p*/*miR-133b* and *miR-1-3p*/*miR-206*, we used our HNSCC miRNA expression signature (GSE184991). All were downregulated (the log_2_ fold-change < −2.0) in HNSCC clinical tissues (Figure 1B). We validated aberrant expression of these miRNAs using the TCGA–HNSC data set. TCGA database analysis showed that all these miRNAs were significantly downregulated in cancer tissues (*n* = 485) compared with normal tissues (*n* = 44) (Figure 1C).

### 3.2. Tumor-Suppressive Functions of miR-1/miR-133 Clustered miRNAs Assessed by Ectopic Expression Assays

To assess the effects of the ectopic expression of *miR-133a-3p*/*miR-133b* and *miR-1-3p*/*miR-206*, functional assays were performed in two HNSCC cell lines (i.e., SAS and Sa3). The results revealed that transfection of *miR-133a-3p*/*miR-133b* and *miR-1-3p*/*miR-206* into HNSCC cell lines significantly suppressed cancer cell proliferation, migration, and invasion (Figure 2A–C). Typical images of cells in migration and invasion assays following *miR-133a-3p*/*miR-133b* and *miR-1-3p*/*miR-206* transfection are shown in Appendix A.

These findings showed tumor suppressive functions of *miR-133a-3p*/*miR-133b* and *miR-1-3p*/*miR-206* in HNSCC.

### 3.3. Screening for Common Oncogenic Targets of Clustered miR-1/miR-133 miRNAs in HNSCC Cells

We focused on target genes coordinately regulated by tumor suppressive clustered miRNAs (*miR-133a-3p*/*miR-133b* and *miR-1-3p*/*miR-206*) that were involved in HNSCC molecular pathogenesis and clinical prognosis.

Our strategy for searching for common putative target genes is shown in Figure 3. Based on the TargetScan Human database (release 7.2), we identified a total of 896 genes that had putative *miR-133a-3p*/*miR-133b*-binding sites in the 3′-UTR, and a total of 711 genes that had putative *miR-1-3p*/*miR-206*-binding sites in the 3′-UTR. Ninety-five of these genes were common putative targets of clustered miRNAs.

Next, we confirmed the expression levels of these genes in HNSCC using TCGA–HNSC data. Among these genes, 28 were upregulated in cancer tissues (*n* = 518) compared to normal tissues (*n* = 44). We further analyzed these 28 genes as candidates for common oncogenic targets (Table 1).

### 3.4. Clinical Significance of miR-1/miR-133 Clustered miRNAs Targets by TCGA Analysis

We investigated the clinical significance of the 28 common putative targets controlled by *miR-133a-3p*/*miR-133b* and *miR-1-3p*/*miR-206* in HNSCC. The expression and 5-year overall survival analysis showed that three genes (*PFN2*, *PSEN1*, and *SYT1*) were significantly upregulated in cancer tissues (Figure 4A), and increased expression levels of each gene were associated with a poorer prognosis in HNSCC patients (log rank test; *p* < 0.05 and false discovery rate < 0.05; Figure 4B and Table 1).

In addition, Cox proportional hazards regression analysis was performed for 5-year overall survival rates, using each gene expression level (i.e., *PFN2, PSEN1,* and *SYT1*), tumor stage, pathological grade, and age as covariates. The multivariate analysis showed that the expression levels of *PFN2* and *PSEN1* were independent prognostic factors (*PFN2*: HR 1.490, *p* < 0.05; *PSEN1*: HR 1.444, *p* < 0,05; Figure 4C). These results suggested that *PFN2* and *PSEN1* were oncogenes related to molecular pathogenesis and clinical prognosis in HNSCC patients.

### 3.5. Direct Control of PFN2 Expression by All Members of the miR-1/miR-133 Clustered miRNAs in HNSCC Cells

First, qRT-PCR was performed to evaluate whether expression of *PFN2* and *PSEN1* was controlled by clustered miRNAs (*miR-133a-3p*/*miR-133b* and *miR-1-3p*/*miR-206*) in HNSCC cells (Figure 5A,B). We found that the expression levels of *PFN2* were significantly suppressed by *miR-133a-3p*/*miR-133b* and *miR-1-3p*/*miR-206* in HNSCC cells (Figure 5A). On the other hand, transfection of *miR-133a-3p*/*miR-133b* reduced the expression level of *PSEN1,* whereas transfection of *miR-1-3p*/*miR-206* did not significantly suppress the expression (Figure 5B). According to these results, we focused on *PFN2* as a common oncogenic target of *miR-133a-3p*/*miR-133b* and *miR-1-3p*/*miR-206* in HNSCC.

Western blotting revealed that the protein levels of PFN2 were reduced by clustered miRNA transfection (Figure 6A). To prove that direct binding between *PFN2* and clustered miRNAs was sequence dependent, a dual-luciferase reporter assay was conducted. The luciferase activity was significantly reduced following co-transfection with *miR-133a-3p*/*miR-133b* and a vector containing the *miR-133a-3p*/*miR-133b*-binding site in the 3′-UTR of *PFN2* (Figure 6B). On the other hand, co-transfection with a vector that lacked the sequence of the *miR-133a-3p*/*miR-133b* site showed no change in luciferase activity (Figure 6B). Co-transfection with *miR-1-3p*/*miR-206* and a vector containing the *miR-1-3p*/*miR-206*-binding site reduced the luciferase activity, but co-transfection with *miR-1-3p*/*miR-206* and a vector lacking the *miR-1-3p*/*miR-206*-binding site did not inhibit luciferase activity (Figure 6C).

### 3.6. Overexpression of PFN2 in HNSCC Clinical Specimens

To confirm expression of the PFN2 protein in HNSCC clinical specimens, immunohistochemical staining was performed. Clinical features of four HNSCC cases used for immunostaining are summarized in Appendix A. Whereas there was almost no PFN2 expression in the normal epithelium, high expression of PFN2 was detected in cancer lesions in HNSCC (Figure 7).

### 3.7. Effects of PFN2 Knockdown on the Proliferation, Migration, and Invasion of HNSCC Cells

To assess the role of *PFN2* as an oncogene in HNSCC cells, functional knockdown assays using siRNA were performed. First, we confirmed the inhibitory effects of siRNA by performing qRT-PCR and Western blotting. Two different siRNAs targeting *PFN2* (i.e., *siPFN2-1* and *siPFN2-2*) were used for this study. The mRNA and protein levels of *PFN2* were significantly inhibited after transfection of siRNAs into HNSCC cell lines (Figure 8A,B).

Then, functional assays using these siRNAs were performed. Knockdown of *PFN2* had little effect on cell proliferation in HNSCC cells (Figure 9A). However, cell migration and invasion were significantly suppressed after transfection of *siPFN2* in SAS and Sa3 cells (Figure 9B,C). Typical images of cells in migration and invasion assays following *siPFN2* transfection are shown in Appendix A.

### 3.8. PFN2-Mediated Molecular Pathways in HNSCC Cells

To identify the molecular pathways involving *PFN2* in HNSCC, we performed gene set enrichment analysis (GSEA) using TCGA–HNSC RNA-seq data. GSEA analysis revealed that “epithelial–mesenchymal transition” was the most enriched pathway in the *PFN2* high expression group (Figure 10 and Table 2). These results suggest that aberrant expression of *PFN2* contributes to the malignant phenotype, including migration and/or invasion of HNSCC, through the epithelial–mesenchymal transition pathway.

## 4. Discussion

Even when various strategies are used to treat HNSCC patients, their prognosis is still poor due to the high rate of recurrence and metastasis [3,7]. Unfortunately, combination therapy with EGFR inhibitor or PD-L1 inhibitor has not achieved satisfactory therapeutic results [37,38]. With the advent of immune checkpoint inhibitors, treatment options for recurrent or metastatic cases have increased, but their efficacy is limited [39,40]. Despite vigorous RNA sequence analysis using samples from patients with HNSCC, the search for therapeutic target molecules for HNSCC has not fully succeeded [41], with a few exceptions [42,43].

We are continuing tumor-suppressive miRNA-based analysis to explore prognostic markers and therapeutic targets in HNSCC [21,23,29,44]. Our previous studies showed that *miR-143* and *miR-145* functioned as tumor-suppressive miRNAs in a wide range of cancers [45,46,47,48]. Notably, they are clustered miRNAs on the human chromosome at region 5q31 [45,46]. Our studies demonstrated that Golgi membrane protein 1 (GOLM1) and hexokinase-2 (HK2) were directly controlled by *miR-143-3p* and *miR-145-5p* in prostate cancer and renal cell carcinoma, respectively [45,46]. Further study showed that the MET proto-oncogene was coordinately regulated by *miR-23b* and *miR-27b* clustered miRNAs in HNSCC cells [49]. Clustered miRNAs regulate one gene through different seed sequences. Continuous and genome-wide analyses of clustered miRNAs are essential to explain the biological implications of clustered miRNAs on the human genome.

Analysis of our original miRNA expression signatures in several cancers, including HNSCC showed that members of the *miR-1*/*133* clustered miRNAs were downregulated in cancer tissues [21,23,50]. Downregulation of each miRNA in the *miR-1*/*miR-133* cluster was confirmed by TCGA–HNSC database analysis. Originally, *miR-1*/*miR-133* clustered miRNAs were discovered through their roles involved in the development of skeletal and cardiac muscles, called “myomiR” [51]. In cancer research, downregulation of *miR-1*/*miR-133* clustered miRNAs were reported in a wide range of cancers, and their functional analyses showed that these miRNAs acted as tumor-suppressive miRNAs [21]. Therefore, the search for oncogenes and oncogenic pathways controlled by each miRNA is being vigorously carried out [52,53,54].

In this study, a total of 28 genes were identified as putative targets of *miR-1*/*miR-133* clustered miRNAs. Among these targets, we identified *PFN2* as a gene directly controlled by all members of *miR-1*/*miR-133* clustered miRNAs in HNSCC cells. Our present analysis showed that aberrantly expressed *PFN2* facilitated cancer cell migration and invasion and was closely involved in the malignant phenotypes of HNSCC.

Profilin is an actin-binding protein that forms an ATP-actin-PFN complex. It recruits monomeric actin to the barbed end of actin filaments and it contributes to elongation [55]. In mammals, profilin is constituted by four members (e.g., PFN1, PFN2, PFN3, and PFN4); PFN1 and PFN2 are the most common types of profilins [56].

Several studies have reported that *PFN2* expression is involved in malignant transformation of cancer cells [57,58,59]. In triple-negative breast cancer (TNBC), high expression of *PFN2* was related to a poorer prognosis (10-year overall survival and relapse-free survival) [58]. In esophageal squamous cell carcinoma (ESCC), PFN2 protein expression was markedly increased gradually from low-grade intraepithelial neoplasia to ESCC, and high expression was positively correlated with the depth of invasion and lymph node metastasis [57]. A recent study reported that *PFN2* is involved in small cell lung cancer metastasis and angiogenesis through exosomes [60].

Transforming growth factor-β (TGF-β) signaling induces cancer cell development and progression [61]. Expression of *PFN2* induced the transactivation of Smad2 and Smad3, and these transmission factors enhanced TGF-β-induced EMT and angiogenesis in lung cancer [59]. *PFN2* overexpression reduced epithelial markers and increased mesenchymal markers in several cancers [57,58]. These results suggest that *PFN2* promotes tumor aggressiveness via EMT. Our present study showed that *PFN2* contributes to the malignant phenotype in HNSCC through the EMT pathway, and these results are consistent with previous reports.

There are other reports that miRNAs control the expression of *PFN2* [62,63,64]. In lung cancer cells, *miR-30a-5p* negatively regulates *PFN2* and inhibited EMT and invasion [62]. In breast cancer, *miR-150-5p* suppressed *PFN2* in a sequence-dependent manner, and the long non-coding RNA *FOXD2* adjacent the opposite strand of RNA1 (*FOXD2-AS1*)/*miR-150-5p*/*PFN2* axis regulated malignancy and tumorigenesis [63]. In osteosarcoma cell lines, *miR-140-5p* repressed *PFN2* [64]. lncRNA *TUG1* was a sponge for *miR-140-5p* to isolate *PFN2*, inducing cell progression and metastasis [64]. Interestingly, these miRNAs were downregulated in HNSCC tissues [65,66,67], and these events might coordinately enhance expression of *PFN2* in HNSCC cells. The finding in this report that the tumor-suppressive clustered miRNAs, *miR-1*/*133* cluster, directly regulated PFN2 in HNSCC cells is attractive and novel. In this study, two cell lines were used to verify the importance of *miR-1*/*miR-133* clustered miRNAs/*PFN2* axes for malignant transformation of HNSCC. In order to test our hypothesis, in vivo (mouse model) experiments are indispensable.

## 5. Conclusions

Analysis of miRNA expression signatures of HNSCC showed that all members of the *miR-1*/*miR-133* miRNA cluster (e.g., *miR-1-3p, miR-206*, *miR-133a-3p*, and *miR-133b*) were frequently downregulated in cancer tissues. TCGA database analysis confirmed that these miRNAs were significantly reduced in cancer tissues. Ectopic expression assays demonstrated that these miRNAs acted as antitumor miRNAs in HNSCC cells. A combination of in silico analyses and luciferase reporter assays revealed that *PFN2* was directly controlled by all members of the *miR-1*/*miR-133* cluster in HNSCC cells. Expression of *PFN2* was closely involved in the prognosis of patients with HNSCC. Moreover, aberrant expression of *PFN2* facilitated cancer cell migration and invasion. Those abilities might be controlled by EMT pathways. Our tumor-suppressive miRNA-based strategy thus provides novel insights, contributing to our overall understanding of the molecular pathogenesis of HNSCC.

## Figures and Tables

**Figure 1 biomedicines-10-00663-f001:**
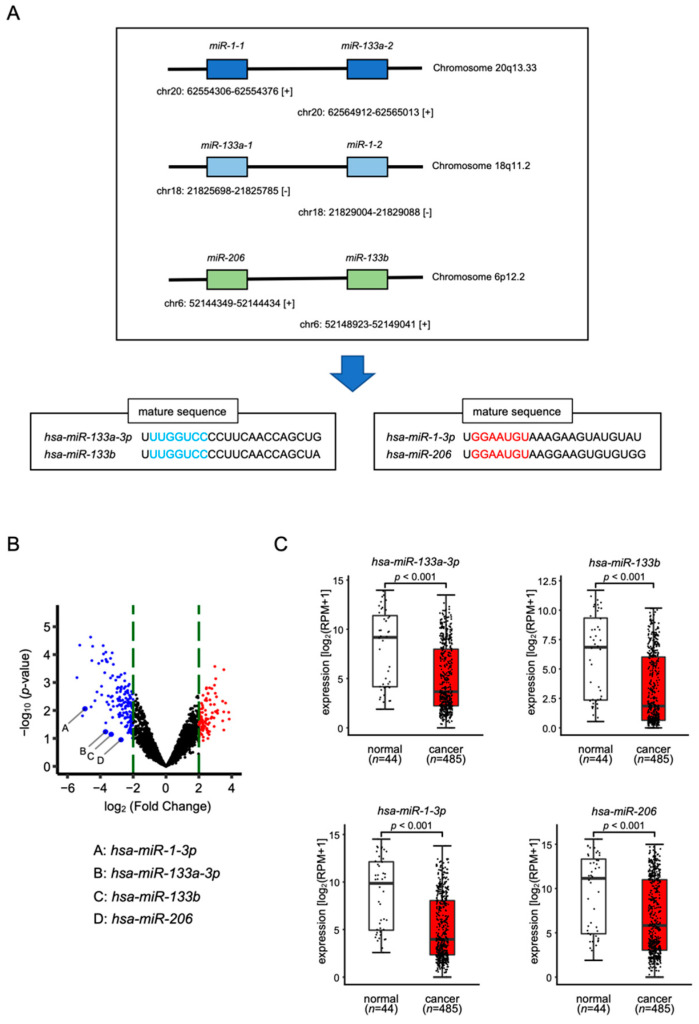
Expression levels of *miR-1*/*miR-133* clustered miRNAs in HNSCC clinical specimens. (**A**) The chromosomal location of each microRNA. Mature seed sequences of *miR-133a-3p*/*miR-133b* were identical. Mature seed sequences of *miR-1-3p*/*miR-206* were identical. (**B**) Volcano plot of the miRNA expression signature determined through small RNA sequencing. The log_2_-fold change (FC) is plotted on the *x*-axis, and the log_10_ (*p*-value) is plotted on the *y*-axis. The blue points represent the downregulated miRNAs with an absolute log_2_ FC < −2.0. The red points represent the downregulated miRNAs with an absolute log_2_ FC > 2.0. (**C**) The expression levels of *miR-133a-3p*/*miR-133b* and *miR-1-3p*/*miR-206* were evaluated in an HNSCC data set from TCGA.

**Figure 2 biomedicines-10-00663-f002:**
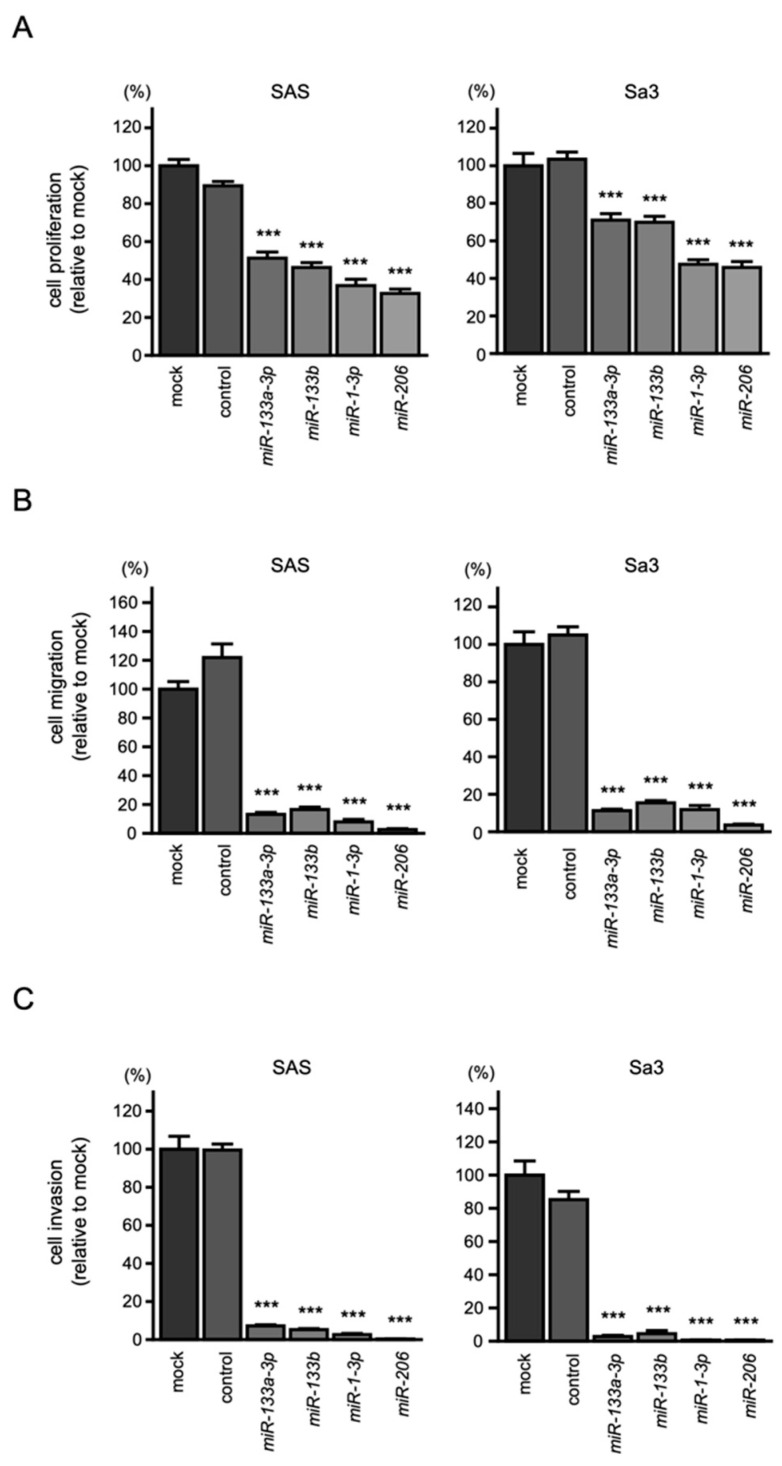
Tumor-suppressive functions of *miR-1*/*miR-133* clustered miRNAs in HNSCC cells, SAS, and Sa3: (**A**) cell proliferation assays; (**B**) cell migration assays; (**C**) cell invasion assays. (*** *p* < 0.001).

**Figure 3 biomedicines-10-00663-f003:**
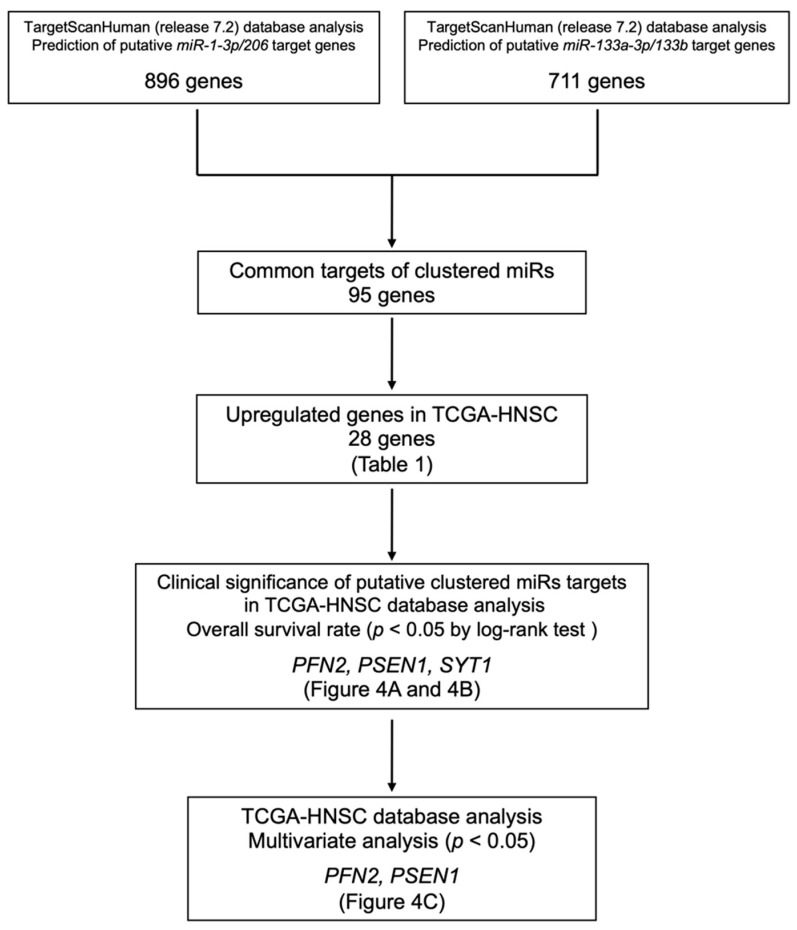
Our strategy of identification of *miR-1*/*miR-133* clustered miRNAs targets in HNSCC cells.

**Figure 4 biomedicines-10-00663-f004:**
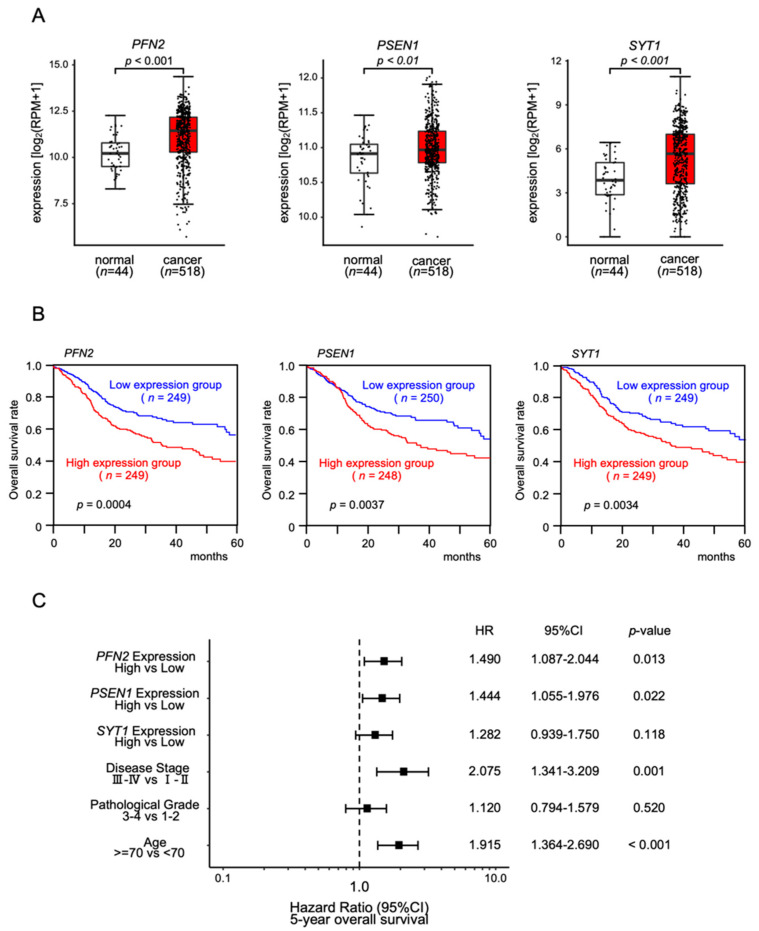
Clinical significance of *PFN2*, *PSEN1,* and *SYT1* in HNSCC clinical specimens determined by TCGA–HNSC analysis: (**A**) comparisons of expression levels of three genes between normal and cancer tissues in TCGA–HNSC; (**B**) Kaplan–Meier curves of the 5 year overall survival frequencies according to the expression of each gene; (**C**) forest plot showing the multivariate analysis results for the three target genes (i.e., *PFN2*, *PSEN1,* and *SYT1*) identified by the analysis of the TCGA–HNSC data (HR: hazard ratio; CI: confidence interval).

**Figure 5 biomedicines-10-00663-f005:**
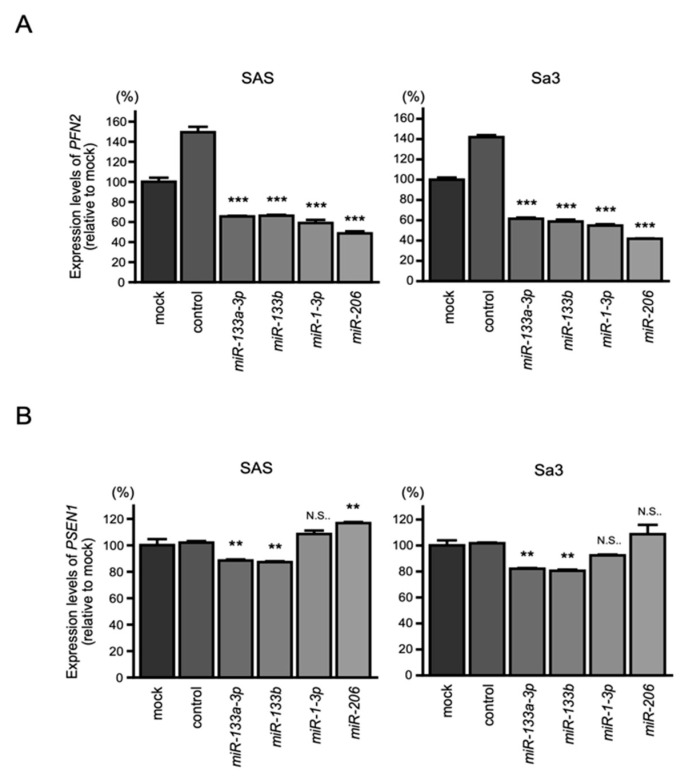
Expression of *PFN2* and *PSEN1* controlled by *miR-1*/*miR-133* clustered miRNAs in HNSCC cells: (**A**) real-time PCR showed significantly reduced expression of *PFN2* mRNA 48 h after transfection of each *miR-133a-3p*/*miR-133b* and *miR-1-3p*/*miR-206*; (**B**) *PSEN1* expression was not repressed by *miR-1-3p*/*miR-206*. (** *p* < 0.01, *** *p* < 0.001, N.S.: not significant).

**Figure 6 biomedicines-10-00663-f006:**
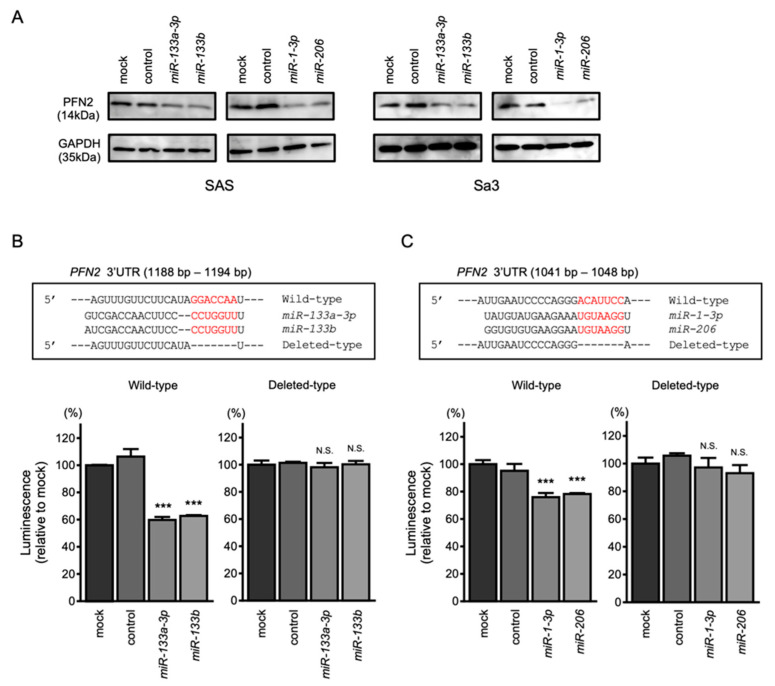
Direct regulation of *PFN2* by *miR-1*/*miR-133* clustered miRNAs in HNSCC cells. (**A**) Western blot of PFN2 protein 48 h after *miR-133a-3p*/*miR-133b* and *miR-1-3p*/*miR-206* transfection of SAS and Sa3 cells. (**B**) The TargetScan database shows that a single putative *miR-133a-3p*/*miR-133b* binding site predicts the 3′-UTR of the *PFN2* sequence (upper panel). Dual-luciferase reporter assays after co-transfection of the wild-type or deleted-type vector and *miR-133a-3p*/*miR-133b* in Sa3 cells (lower panel). (**C**) The TargetScan database shows that a single putative *miR-1-3p*/*miR-206* binding site predicts the 3′-UTR in the *PFN2* sequence (upper panel). Dual-luciferase reporter assays after co-transfection of the wild-type vector or deleted-type vector and *miR-133a-3p*/*miR-133b* in Sa3 cells (lower panel). (*** *p* < 0.001, N.S.: not significant).

**Figure 7 biomedicines-10-00663-f007:**
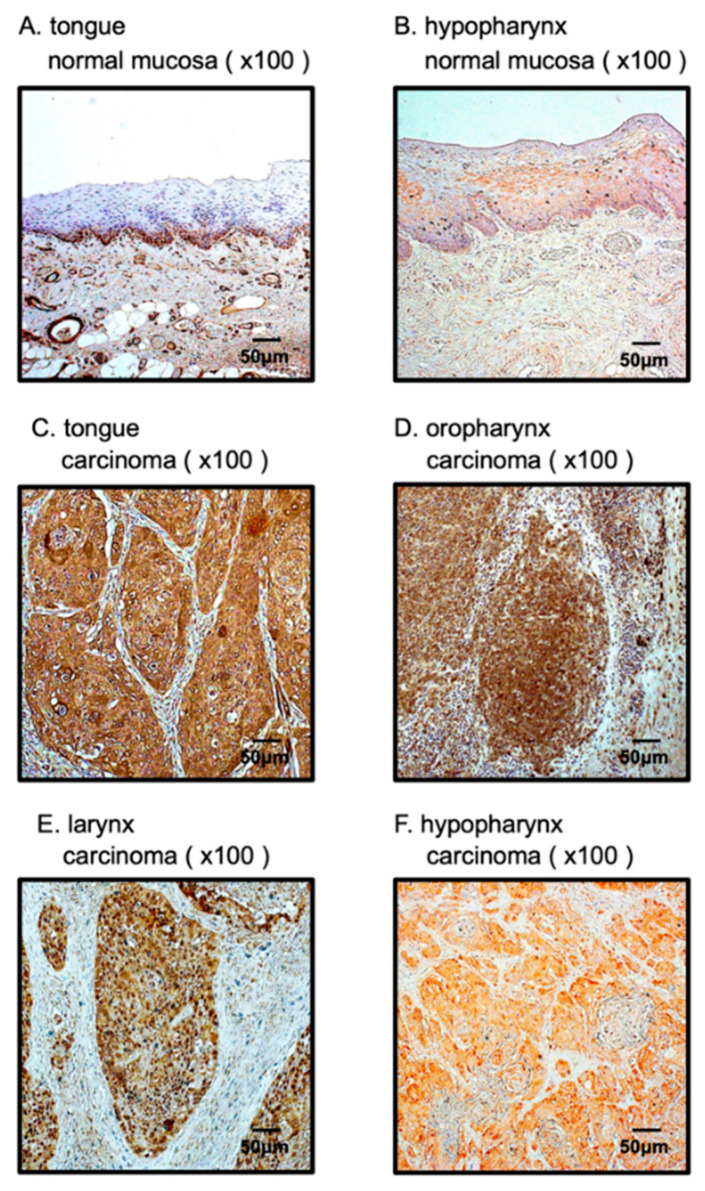
Overexpression of PFN2 in HNSCC clinical specimens. (**A**,**B**) Weak expression was detected in the normal mucosa. (**C**–**F**) High expression of PFN2 was detected in the nuclei and/or cytoplasm of HNSCC cancer cells.

**Figure 8 biomedicines-10-00663-f008:**
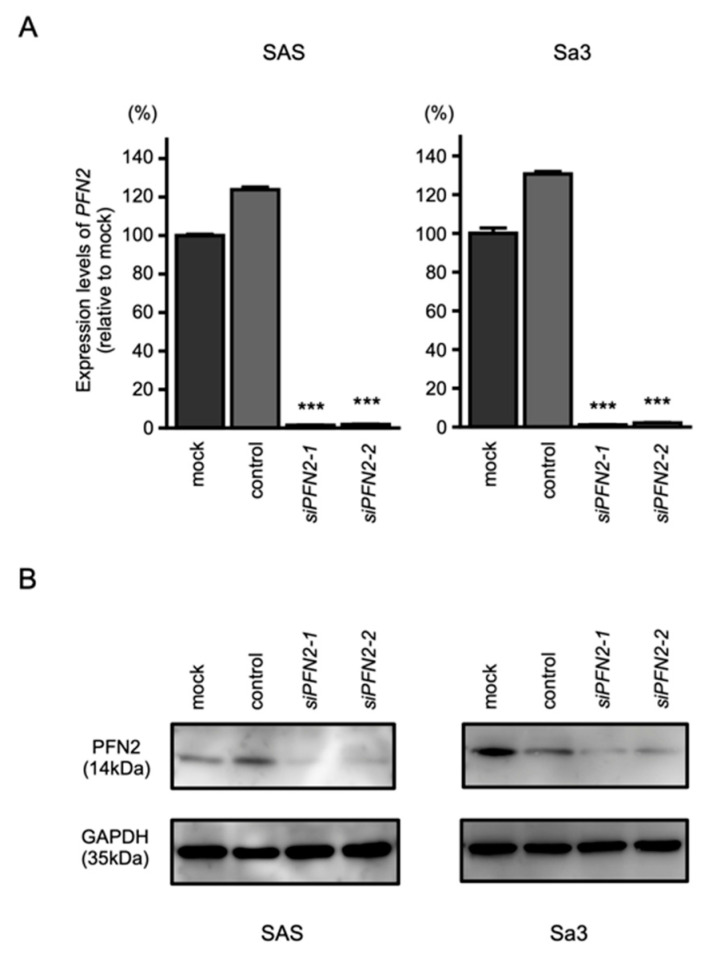
Knockdown efficiencies of siRNAs targeting *PNF2* in HNSCC cell lines. Knockdown efficiencies of *PFN2* expression by *siPFN2-1* and *siPFN2-2* were evaluated by real-time PCR (**A**) and Western blotting (**B**) in SAS and Sa3 cells. Expression data for *PFN2* (mRNA) and PFN2 (protein) were collected 48 h after siRNAs transfection. (*** *p* < 0.001).

**Figure 9 biomedicines-10-00663-f009:**
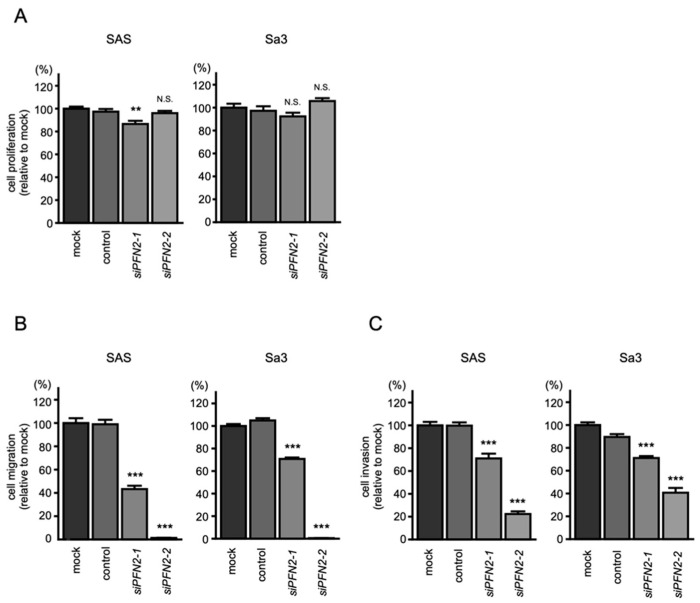
Functional assays after knockdown of *PFN2* in HNSCC cell lines (SAS and Sa3): (**A**) cell proliferation assays; (**B**) cell migration assays; (**C**) cell invasion assays. (** *p* < 0.01, *** *p* < 0.001, N.S.: not significant).

**Figure 10 biomedicines-10-00663-f010:**
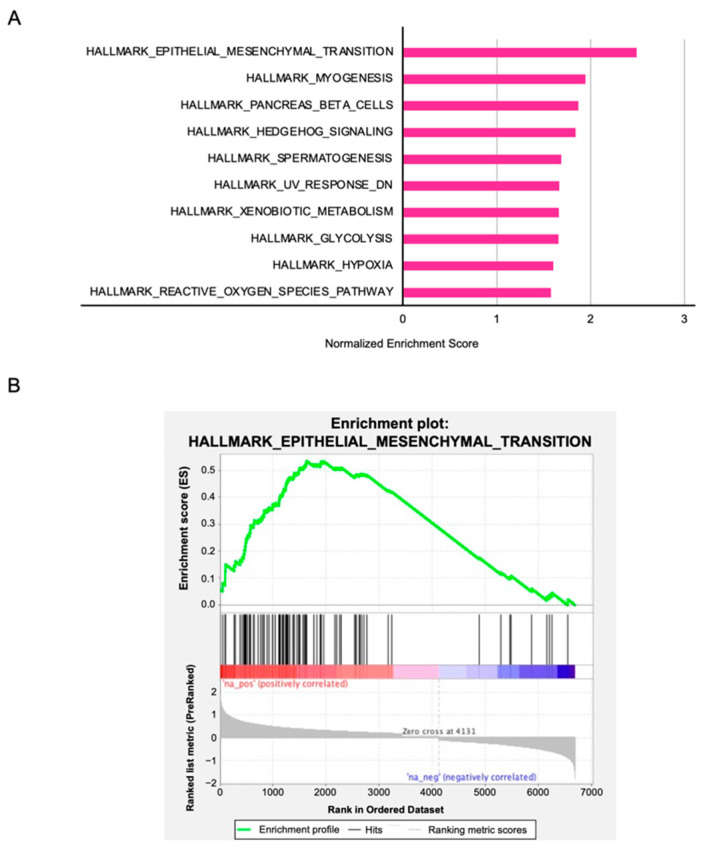
*PFN2*-mediated pathways identified by gene set enrichment analysis: (**A**) the top 10 enriched gene sets in the high *PFN2* expression group; (**B**) enrichment plot of “epithelial–mesenchymal transition”.

**Table 1 biomedicines-10-00663-t001:** Twenty-eight candidate target genes regulated by both *miR-133a-3p*/*miR-133b* and *miR-1-3p*/*miR-206*.

Entrez Gene ID	Gene Symbol	Gene Name	5 y OS *p*-Value (Log-Rank Test)	FDR (Benjamini–Hochberg)
27	*ABL2*	c-abl oncogene 2, non-receptor tyrosine kinase	0.155	0.426
29956	*CERS2(LASS2)*	ceramide synthase 2	0.353	0.618
54805	*CNNM2*	cyclin M2	0.792	0.822
23603	*CORO1C*	coronin, actin binding protein, 1C	0.099	0.358
57089	*ENTPD7*	ectonucleoside triphosphate diphosphohydrolase 7	0.073	0.354
23197	*FAF2*	Fas associated factor family member 2	0.406	0.661
2729	*GCLC*	glutamate-cysteine ligase, catalytic subunit	0.710	0.764
23349	*KIAA1045(PHF24)*	KIAA1045(PHD finger protein 24)	0.110	0.358
55243	*KIRREL*	kin of IRRE like (Drosophila)	0.638	0.733
3927	*LASP1*	LIM and SH3 protein 1	0.051	0.354
27253	*PCDH17*	protocadherin 17	0.167	0.426
5150	*PDE7A*	phosphodiesterase 7A	0.655	0.733
5217	*PFN2*	profilin 2	0.000	0.011
5663	*PSEN1*	presenilin 1	0.004	0.035
5725	*PTBP1*	polypyrimidine tract binding protein 1	0.621	0.733
5757	*PTMA*	prothymosin, alpha	0.330	0.616
5814	*PURB*	purine-rich element binding protein B	0.229	0.492
285590	*SH3PXD2B*	SH3 and PX domains 2B	0.552	0.733
55186	*SLC25A36*	solute carrier family 25 (pyrimidine nucleotide carrier), member 36	0.443	0.661
6546	*SLC8A1*	solute carrier family 8 (sodium/calcium exchanger), member 1	0.919	0.919
6857	*SYT1*	synaptotagmin I	0.003	0.035
8407	*TAGLN2*	transgelin 2	0.626	0.733
7030	*TFE3*	transcription factor binding to IGHM enhancer 3	0.472	0.661
79183	*TTPAL*	tocopherol (alpha) transfer protein-like	0.115	0.358
26100	*WIPI2*	WD repeat domain, phosphoinositide interacting 2	0.076	0.354
7525	*YES1*	v-yes-1 Yamaguchi sarcoma viral oncogene homolog 1	0.205	0.478
56829	*ZC3HAV1*	zinc finger CCCH-type, antiviral 1	0.269	0.538
55609	*ZNF280C*	zinc finger protein 280C	0.455	0.661

5 y OS: 5-year overall survival rates; FDR: false discovery rate.

**Table 2 biomedicines-10-00663-t002:** The top 10 enriched gene sets in the high *PFN2* expression group.

Name	Normalized Enrichment Score	FDR *q*-Value
HALLMARK_EPITHELIAL_MESENCHYMAL_TRANSITION	2.490	*q* < 0.001
HALLMARK_MYOGENESIS	1.943	0.008
HALLMARK_PANCREAS_BETA_CELLS	1.870	0.012
HALLMARK_HEDGEHOG_SIGNALING	1.840	0.012
HALLMARK_SPERMATOGENESIS	1.685	0.047
HALLMARK_UV_RESPONSE_DN	1.664	0.046
HALLMARK_XENOBIOTIC_METABOLISM	1.662	0.040
HALLMARK_GLYCOLYSIS	1.658	0.036
HALLMARK_HYPOXIA	1.600	0.053
HALLMARK_REACTIVE_OXYGEN_SPECIES_PATHWAY	1.577	0.058

## Data Availability

Our expression data were deposited in the GEO database (accession number: GSE189290).

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
