# Peer review of "Impact of miR-1/miR-133 Clustered miRNAs: PFN2 Facilitates Malignant Phenotypes in Head and Neck Squamous Cell Carcinoma"

_biomedicines, 2022, doi:10.3390/biomedicines10030663_

Round 1
Reviewer 1 Report
This is an interesting report about the impact of miR-1/miR-133 clustered miRNAs, PFN2 facilitates 2 malignant phenotypes in head and neck squamous cell carcinoma.
-However, above all in the Discussion, I ound not so much important translational points about this research. Accordingly the authorus should introduce a paragraph where they explain which will be the translational effects in the daily clinical practice regarding their research.
-Add in the discussion that in the head/neck region melanoma and non melanoma skin cancers may show different cliical outcome regarding their specific position.
- Finally, please speculate and compare your results with the possibility to introduce also the use of extracellular vescicles in the daily clinical practice for the followup of these patients, as also reported recently in this article, that the authors can add to their references to better higlight a more translational point of view of their research "The Fatty Acid and Protein Profiles of Circulating CD81-Positive Small Extracellular Vesicles Are Associated with Disease Stage in Melanoma Patients. Cancers (Basel). 2021 Aug 18;13(16):4157. doi: 10.3390/cancers13164157. PMID: 34439311; PMCID: PMC8392159."
Author Response
This is an interesting report about the impact of miR-1/miR-133 clustered miRNAs, PFN2 facilitates 2 malignant phenotypes in head and neck squamous cell carcinoma.
Comment-1: However, above all in the Discussion, I ound not so much important translational points about this research. Accordingly the authorus should introduce a paragraph where they explain which will be the translational effects in the daily clinical practice regarding their research.
Response: As suggested by the reviewer’s comment, I have referred to some articles on the clinical deployment of microRNAs and added the following new text in Introduction.
Recent advances in nanotechnology have led to the development of drug delivery systems that deliver various drugs to target cancer cells [14]. Many attempts to use miRNAs as pharmaceuticals have been reported so far [15,16]. The advantage of miRNAs as drugs is that one type of miRNA has the potential to control many target genes [15]. Recently, exosomes have been attracting attention as a drug delivery system. Exosomes are a type of cell-derived vesicles characterized as extracellular vesicles. Of particular note, some miRNAs are contained within exosomes, and migrate between cells through exosomes [14,17]. The development of new therapies that embed tumor-suppressive miRNAs in exosomes and deliver them to cancer cells is very attractive.
Comment-2: Add in the discussion that in the head/neck region melanoma and non melanoma skin cancers may show different cliical outcome regarding their specific position.
Response: I would like to thank the reviewer for comment on the paper. This study is a paper in which microRNA analysis was performed for squamous cell carcinoma of the head and neck. Melanoma of head and neck cancer is outside the scope of this analysis. I don't have any data on melanoma. Also, I don't have enough knowledge of melanoma, so I would like to avoid mentioning melanoma in this paper. I appreciate your understanding.
Comment-3: Finally, please speculate and compare your results with the possibility to introduce also the use of extracellular vescicles in the daily clinical practice for the followup of these patients, as also reported recently in this article, that the authors can add to their references to better higlight a more translational point of view of their research "The Fatty Acid and Protein Profiles of Circulating CD81-Positive Small Extracellular Vesicles Are Associated with Disease Stage in Melanoma Patients. Cancers (Basel). 2021 Aug 18;13(16):4157. doi: 10.3390/cancers13164157. PMID: 34439311; PMCID: PMC8392159."
Response: I would like to thank the reviewer for comment on the paper. I would also like to thank you for introducing us to an excellent paper on melanoma. However, this paper is not suitable for consideration of our paper because we focused on HNSCC. I appreciate your understanding.
Thank you for your constructive comments and suggestions.
Reviewer 2 Report
The submitted manuscript entitled “Impact of miR-1/miR-133 clustered miRNAs: PFN2 facilitates malignant phenotypes in head and neck squamous cell carcinoma” authored by Dr. Asai and others is an interesting finding. Overall, by combining bioinformatic analysis and experimental evidence, the data are suitable to support the hypothesis. I have a few questions before considering publishing.
- The language is required to be improved by a scientific institution, since many sentences are difficult to understand.
- Based on migration and invasion assays, they appear to have significant effects on these phenotypes. Did the authors have any data to reveal which pathways are involved?
- Are there normal cell lines with these mi expressions? Is there any effect on the health of these cell lines if the same procedure is applied?
- Statistical analysis doesn't seem very professional. The authors are required to re-analyze all data such as *<0.5, **<0.01, etc.
- In Figure 7, the quality of the IHC is very poor and there appears to be heavy background signal. High-resolution images should be included in new submissions.
- The two tested cell lines should be used to establish in vivo mouse models to evaluate migration, invasion, and tumor growth under different treatment conditions.
Author Response
Reviewer #2
The submitted manuscript entitled “Impact of miR-1/miR-133 clustered miRNAs: PFN2 facilitates malignant phenotypes in head and neck squamous cell carcinoma” authored by Dr. Asai and others is an interesting finding. Overall, by combining bioinformatic analysis and experimental evidence, the data are suitable to support the hypothesis. I have a few questions before considering publishing.
Comment-1: The language is required to be improved by a scientific institution, since many sentences are difficult to understand.
Response: I would like to thank the reviewer for comment on the paper. I would like to hear the opinions of the editors and make a decision. If necessary, request proofreading in English again.
Comment-2: Based on migration and invasion assays, they appear to have significant effects on these phenotypes. Did the authors have any data to reveal which pathways are involved?
Response: Reviewer's comment is an important point to this paper. To investigate miR-1/miR-133 clustered miRNAs/PFN2 axis involved cancer pathways, we performed GESA for analysis (Results 3.8: PFN2-mediated molecular pathways in HNSCC cells, Figure 10). Most enriched pathway is EMT, I mentioned this in the results (3.8).
Comment-3: Are there normal cell lines with these mi expressions? Is there any effect on the health of these cell lines if the same procedure is applied?
Response: The reviewer's point is judged to be an important analysis item. I investigated the expression of miR-1, miR-206, miR-133a, and miR-133b in other cancer cell lines, and non-cancerous cell lines, to performed loss-of-function analysis of these miRNAs using miRNA inhibitors. However, no cell line was found in which the expression of these microRNAs was sufficiently maintained. The reviewer's suggestions are appealing, but difficult to analyze this time around. I’m grateful for your understanding.
Comment-4: Statistical analysis doesn't seem very professional. The authors are required to re-analyze all data such as *<0.5, **<0.01, etc.
Response: As suggested by the reviewer’s comment, I corrected the description of Statistical analysis as below, and corrected the significance differences in Figure 2, 5,6,8 and 9.
2.11. Statistical analysis
JMP Pro 15 (SAS Institute Inc., Cary, NC, USA) was used for statistical analyses. Comparisons between the two groups were assessed by Welch’s t-test. Differences between multiple groups were assessed by Dunnett’s test compared to control group. A p-value < 0.05 was considered statistically significant. Significant differences within the figures are expressed as follows: *: p < 0.05, **: p < 0.01, ***: p < 0.001, N.S.: not significant. Quantitative data are presented as the means and standard errors.
Comment-5: In Figure 7, the quality of the IHC is very poor and there appears to be heavy background signal. High-resolution images should be included in new submissions.
Response: As suggested by the reviewer’s comment, I took the pictures of immunostaining again.
Comment-6: The two tested cell lines should be used to establish in vivo mouse models to evaluate migration, invasion, and tumor growth under different treatment conditions.
Response: I’m aware that the reviewers' point is an important for our research (cancer research). However, analysis using mouse model takes a lot of time and research costs. Please understand that this article skips the analysis of the mouse model. In response to the reviewers' suggestion, I mention the problems with our analysis as follows in Discussion.
In this study, two cell lines were used to verify the importance of miR-1/miR-133 clustered miRNAs/PFN2 axes for malignant transformation of HNSCC. In order to test our hypothesis, in vivo (mouse model) experiments are indispensable.
Thank you for your constructive comments and suggestions.
Round 2
Reviewer 2 Report
The in vivo data should be included. Thanks!